# Who Is Best Placed to Support Cyber Responsibilized UK Parents?

**DOI:** 10.3390/children10071130

**Published:** 2023-06-29

**Authors:** Suzanne Prior, Karen Renaud

**Affiliations:** 1School of Design and Informatics, Abertay University, Dundee DD1 1HG, UK; s.prior@abertay.ac.uk; 2Department of Computer and Information Sciences, University of Strathclyde, Glasgow G1 1XQ, UK; 3School of Computer Science, University of South Africa, Pretoria 0003, South Africa; 4Department of Information Systems, Rhodes University, Grahamstown 6140, South Africa

**Keywords:** parents, cybersecurity, responsibilization

## Abstract

The UK government responsibilizes its citizens when it comes to their cyber security, as do other countries. Governments provide excellent advice online, but do not provide any other direct support. Responsibilization is viable when: (1) risk management activities require only ubiquitous skills, (2) a failure to manage the risk does not affect others in the person’s community. Cybersecurity fails on both counts. Consider that parents and carers are effectively being responsibilized to educate their children about cybersecurity, given that young children cannot be expected to consult and act upon government advice. Previous research suggests that UK parents embrace this responsibility but need help in keeping up to date with cybersecurity ‘best practice’. In this paper, we consider a number of possible sources of parental advice, and conclude that support workers would be best placed to support parents in this domain. We then carried out a study to gauge the acceptability of this source of help. We find that parents would be willing to accept advice from this source, and suggest that cybersecurity academics be recruited to train support workers to ensure that they have current ‘best practice’ cybersecurity knowledge to impart to parents.

## 1. Introduction

Global citizens increasingly inhabit a digital-first world, with many services being offered online as a preferred option [1]. In 2020, UK citizens spent an average of 4 h per day online [2]. One in three Internet users were children in 2015 [3], and the Institution of Engineering and Technology (IET) reported in 2022 that children now spend more time online than in the real world [4], likely a consequence of the pandemic lockdowns [5,6,7,8,9].

Criminals and perverts operate in the online world to pursue their own nefarious aims. Cybercriminals target individuals, businesses and entire countries [10]. Children, too, are likely to be targeted and might actually be at greater risk than their parents [11]. As such, children should know how to keep themselves both safe and secure online [12], as they learn to do in the physical world. Safety and security are substantially different. ***Cybersafety*** is related to preventing harms resulting from online *content* (seeing adult content), *contact* (being contacted by unknown adults) and *conduct* (misbehaving online) [13]. ***Cybersecurity*** can simply be defined as “*the protection of cyber-systems against cyber-threats*” ([14], p. 29). As such, cybersafety is related to protecting the child’s person and well being (including preventing cyberbullying), while cybersecurity is related to protecting the child’s devices and information.

Parents routinely teach their children how to keep themselves safe in the physical world [15]. Interestingly, Statista published data which demonstrates that UK parents are also the primary source of ***cybersafety*** information for their children throughout their childhood [16]. This study found that 91% of children aged 12–15 stated that their parents were a source of cybersafety-related information. Contrast this with the percentage who stated that they received cybersafety information from their friends, which stood at only 14%. Teachers were the second most common source of information, but still only accounted for 66–73% of children. Moreover, Smahel et al. [17] reported that parents were the main source of online-related support for children and young people. There is evidence that parental mediation can indeed support young learners in managing online risk [18].

In this paper, we consider how children can be taught about cybersecurity, and do not address cybersafety, while not denying its importance in this space. Now, consider that UK citizens, as with other neo-liberalised nations, are responsibilized to take care of their own cybersecurity [19]. A subset of citizens are parents, and they, too, are responsibilized not only to take care of their own cybersecurity, but also the cybersecurity of their children [20,21].

Previous research found that parents were happy to accept the responsibility to educate their children about cybersecurity [22,23]. However, the researchers also found that UK parents’ cybersecurity knowledge was often out of date. This points to the need for UK parents to be supported and empowered if they are to be held responsible for their children’s cybersecurity. It might seem obvious that the government, who is responsibilizing the parents, ought to ensure that parents are able to embrace and fulfil this responsibility. However, trust in the UK government is currently very low (with only 35% of the UK population trusting the UK government [24]). This low percentage has been confirmed by the 2023 Edelman Trust Barometer [25].

In essence, we now have a situation where parents need help to bring their cyber knowledge up to date, but they also do not trust the very entity who is able to provide them with this cybersecurity assistance and advice [26]. In this paper, we report on an alternative way to empower UK parents in this respect, to help them to act upon and fulfil their cyber responsibilities towards their children. We carried out a Q-methodology study to gauge which source of correct and trustworthy advice would be deemed acceptable to UK parents.

Section 2 reviews the related research. Section 3 proposes a way to provide UK parents with more support in educating their children about cybersecurity. Section 3.1 then outlines the study we carried out to determine the acceptability of our proposed intervention to UK parents. Section 4 discusses the results and considers the research implications of our findings and mentions the limitations of this investigation. Section 5 concludes.

## 2. Related Work

### 2.1. Responsibilization

Responsibilization theory [27] revolves around the concept of assigning responsibility to individuals and influencing them to embrace those responsibilities. This theory emphasises the transformation of individuals into self-reflexive and self-directed agents, capable of taking charge of various aspects of their lives without relying heavily on government support. The aim of responsibilization is to promote self-reliance and reduce the burden on government services. According to Pellandini-Simányi and Conte [28], responsibilization encompasses both the assignment of responsibility to citizens and the influence of social and cultural factors that persuade individuals to accept and fulfill those responsibilities. The theory aims to create a society of empowered individuals who actively participate in decision-making processes and take ownership of their actions. Effective implementation of responsibilization theory requires two key factors.

Firstly, citizens must be willing and able to carry out their cyber-related responsibilities. It is crucial for them to possess the necessary competence to embrace their responsibilities fully. This involves equipping individuals with the knowledge, skills, and resources required to fulfil their assigned roles. Secondly, it is crucial to ensure that the failure of individuals to act upon their responsibilities does not harm others. This aspect highlights the importance of balancing individual autonomy and collective well-being within the framework of responsibilization. By examining these dimensions, responsibilization theory provides insights into how governments and societies can promote a culture of self-sufficiency while maintaining social cohesion and responsibility.

Given the lack of cybersecurity knowledge demonstrated by UK parents [22,29], it is unlikely that the first requirement can be met. UK parents’ inability to fulfil their state-assigned and willingly accepted responsibility to educate their children about cybersecurity is likely to result in the unwitting dissemination of incorrect and outdated advice. This, in turn, will result in the widening of the cybersecurity divide [30]. Parents might not be aware of this situation, and even if they are aware, might well choose not to confront it or actively seek to improve their cybersecurity knowledge. If they realise that they are in this quandary, they might well consult external support or resources to get advice.

### 2.2. Children’s Cybersecurity Education

There is widespread acknowledgement that it is important to educate children about cybersecurity [31,32,33,34,35,36,37]. Throughout the UK, cybersecurity is included in the curriculum for children throughout their time at school. However, the extent to which it is covered varies by individual nation [38]. In addition, no curriculum in England fully covers the basic topics defined by the UK government as being necessary for good personal cybersecurity [39].

Some authors have commented on the fact that while awareness is high, or improved by lessons, this does not necessarily convert to secure behaviours [40,41]. This suggests that efforts at school need to be augmented at home, so that educational efforts are reinforced by parents, as is common for other educational domains [42].

Quayyum [43] argues for the significant role parents play in in cybersecurity education. Therefore, it falls to UK parents to educate their children. Previous research has shown that UK parents embrace this responsibility [22]. They might well consider this to be part of the activities they carry out to consider themselves to be ‘good parents’. The danger in relying upon parents to fulfil this role without external support is that their level of knowledge cannot be guaranteed to be sufficient enough to perform this task. Indeed, recent research has suggested it is poor [22], in line with cybersecurity knowledge across the UK population [44].

A variety of cybersecurity resources are freely available [45]. However, there are issues with many of these, in particular books, which have often been found to contain out-of-date or incorrect guidance [46]. In addition, even when children are able to engage independently with the many online resources [38], they still require input from adults to explain how to apply the principles. That being so, we cannot feasibly expect children to teach themselves the correct principles.

### 2.3. Empowering UK Parents with Cyber Advice

While it is crucial that efforts continue to be made in addressing and improving children’s knowledge, similar efforts must be invested into improving parents’ knowledge, given their responsibility for teaching their children about cybersecurity best practice.

At present, it would appear that parents do not possess sufficient up-to-date knowledge to be responsible for managing the cybersecurity education of their children, nor do they consult the most reliable sources to obtain such knowledge [22]. Recent government campaigns have had little impact (e.g., using passphrases instead of complex passwords) [47].

Even so, parents do demonstrate a willingness to accept advice from educational authorities and cybersecurity academics. This offers an opportunity for intervening to provide more support to responsibilized UK parents. Advice sources can be judged on two dimensions: (1) whether they provide correct advice, and (2) whether they are trusted [48,49]. The first is crucial because if someone gets the wrong or outdated cybersecurity advice, at some point they are likely to be confronted by the correct advice. When that happens, they have to unlearn the previous information, which is challenging to do [50,51]. The second is equally important because trust levels influence behaviours [52]. Let us now consider each of the possible sources of advice as shown in Figure 1.

#### 2.3.1. Government as Source

While responsibilization is a government strategy, and embraced by parents, it is clearly a challenge for governments to support parents in this respect due to low trust levels [24,25,26]. The respondents in [22] were unwilling to trust cybersecurity-related advice issued by the UK government, possibly due to recent events in the UK [53,54].

#### 2.3.2. Teachers as Source

Parents were willing to accept advice from education professionals and cybersecurity academics [22]. Moreover, Ipsos MORI surveys revealed that teachers remained a trusted profession in the UK [55,56]. However, teachers themselves also appear to struggle to keep up with the latest cybersecurity knowledge [12,57,58]. Moreover, teachers’ workloads are such that it is unlikely they have the bandwidth to keep up with the latest cybersecurity developments or to provide support to parents [59]. Cybersecurity education is indeed included in primary school curricula, but the topics and approaches to teaching these vary, depending on the educational authority [38]. Moreover, childhood deprivation has a deleterious impact on how well children learn these principles at school [30].

#### 2.3.3. Family and Friends as Source

Muir and Joinson [23] reported that many people relied on family and friends for cybersecurity advice. In another study, it emerged that a third of parents sought cybersecurity advice from their own friends and family [22]. However, given the general lack of cybersecurity knowledge across the UK population [22], this might well lead to incorrect advice spreading throughout the community, which may further widen the cyber divide discussed in [30].

#### 2.3.4. Children’s Books as Source

The UK has societies that encourage the buying of books for children [60]. Queen Camilla and the Princess of Wales promote the reading of books to children [61]. However, children’s books do not provide up-to-date cyber security principles [46]. This is unsurprising since the advice responds to dynamic and ever-changing cyber threats, and books, in print format, cannot possibly change as quickly.

#### 2.3.5. Cybersecurity Academics as Source

UK academics are generally trusted by the public [62]. The pandemic might easily have dented this trust [63] but luckily, the British Academy’s recent study found that academics are still widely trusted [64]. Cybersecurity academics work with psychologists and computer scientists to understand how to deliver cybersecurity guidance to the public. Cybersecurity academics can also be expected to give correct advice, given that this is their primary focus.

However, research related to empowering parents is still rather sparse. For example, Al-Naser et al. [65] revealed a lack of information for children aged six years or below, when it comes to smart device cybersecurity. Prior and Renaud [45] developed an age-appropriate approach to educating children about password best-practice, which was intended to help both teachers and parents, and to fill the knowledge gap that exists with respect to password ‘best practice’.

#### 2.3.6. Online Advice as Source

Google determined, in a 2019 study, that people tended to trust their search results [66]. Moreover, [22] found that people preferred to rely on results returned by a search engine in terms of finding advice on cybersecurity matters, as opposed to other sources of information. However, there is a great deal of variety in offered advice and Renaud and Prior [12] found that many online sources provided incorrect and out-of-date advice, unlike the advice provided by the UK’s National Cybersecurity Centre [67].

For example, at present the BBC (the UK public broadcasting service) website for children (CBBC) [68] requires the use of lower case, upper case, digit, special character (LUDS) when creating a password. This is outdated and no longer ‘best practice’ [67].

The CBBC site also asks the user to set a username. While it does not require this to be the child’s actual name, it also does not disallow this. Furthermore, if the child forgets their password, it cannot be reset and a new account must be created. This may encourage the use of very weak passwords.

#### 2.3.7. Technology Providers as Source

There is certainly an opportunity for technology providers and companies to be proactive in protecting children’s cybersecurity. This should be routine throughout all reputable online services, which is not the case at present.

The UK Government is working on ‘Secure by Design’ legislation called Product Security and Telecommunications Infrastructure (PSTI) Act. This, if realised, could give technology providers a role to play in terms of more secure architectures for computer hardware. However, the idea that they could play a role in providing advice to UK citizens does not appear to have been envisaged. At the moment, this group does not seem to be widely consulted for cybersecurity advice [22].

#### 2.3.8. Support Workers as Source

Support workers or family liaison workers enjoy a great deal of trust in the UK [69]. These are are professionals who provide support, guidance, and assistance to families in challenging circumstances or during significant events. These individuals are typically employed within various sectors, such as law enforcement agencies, social services, or educational institutions, to act as a vital link between families and the relevant organisations or authorities.

Support workers facilitate information exchange, coordinate services, and ensure that families have access to the necessary support networks and play a crucial role in promoting collaboration and cooperation between families and the organisations.

This has the potential to bridge the cybersecurity knowledge gap. Within UK schools, these members of staff work primarily with ‘at risk’ families, supporting them in engaging with the schools, with financial challenges or supporting their children through educational difficulties [70]. They also work with the school community as a whole.

However, they do not necessarily have current cybersecurity knowledge since their training is usually health-related. This study set out to identify the most acceptable source of information. The next step would be to explore the logistics of delivering cybersecurity advice in this way—support workers are often overloaded [71]. Unless this issue is addressed, they will not be in a position to take on this additional duty.

### 2.4. In Summary

We have reviewed the literature and considered all usual sources of cyber advice and found issues with all of them. However, it is worth noting that the UK’s National Cyber Security Strategy [72] includes the following statement: “*Civil society organisations and community groups also play a major role supporting people to understand and protect themselves from cyber risks. Many charities, for example, provide targeted support, advice and awareness-raising to vulnerable groups*”. This suggests that we should go beyond traditional sources in empowering UK parents. Table 1 summarises the discussion, showing the trustworthiness of the different cybersecurity advice sources, based on Redmiles et al.’s [49] findings that ‘*participants evaluated digital-security advice based on the trustworthiness of the advice source’*.

## 3. Proposal

It is crucial that parents’ knowledge is improved, especially parents who are at the greatest risk of exclusion from traditional educational sources i.e., those with lower levels of education who have less or incorrect cybersecurity knowledge. Their children are likely also to have have lower levels of knowledge, given the findings of other studies.

One group of professionals already working with these families are family support or liaison workers. These professionals already take on a variety of roles, and, as such, they would also require support in order to be able to include cybersecurity as part of the guidance they offer to families. It is important to acknowledge that this should be considered a long-term endeavour. The challenges of achieving this include:

(1) Ensuring that there are enough support workers, where the UK currently has a shortage (https://teach-now.co.uk/tackling-the-teaching-assistant-shortage/, accessed on 24 June 2023). Moreover, recruiting those who would be willing to undertake their current duties as well as their new cybersecurity advice-giving duties;

(2) Linked to (1), recruiting enough new trainees to ensure that the number of support workers stay at a constant level;

(3) Obtaining funding from government for this endeavour [73], especially post-pandemic when it is more likely for funding to be cut [74];

(4) Cybersecurity would have to be added to a support worker training framework such as the UK’s National Health Service’s AHP Support Worker Competency, Education and Career Development Framework (https://www.hee.nhs.uk/our-work/allied-health-professions/enable-workforce/developing-role-ahp-support-workers/ahp-support-worker-competency-education-career-development, accessed on 24 June 2023). It takes time for this kind of change to be approved at all levels;

(5) These workers are poorly paid at present [73], and unless this is remedied, it is unlikely that they would be willing to take on extra duties;

(6) Managing parental expectations is crucial [73]. It would be all too easy for them to be asked, by parents and the school, to fix computers or remove malware. Moreover, the school might want them to teach the children about cyber, which would make their role unsustainable. Their role would have to be clearly defined in their job statement to protect them. It would have to be made clear that their role is **only** to provide correct and trustworthy advice;

(7) Finally, looking to the future, it is likely that there would be calls for certifications of these workers in their cyber advice role. If extra training could help to address the current lack of career progression opportunities and the current lack of respect for these workers [73], this could be a positive, but only if this is something they would want to do.

Academics working in cybersecurity could work directly with these professionals, training them in the most up-to-date cybersecurity knowledge. They would then be able to communicate this to the families they assist as part of their work in engendering good education and health practices (see Figure 2).

Resources for families should be designed considering a variety of ethnic groups and languages. By working with various groups and language specialists, academics could help to produce resources for use for all parents. This would require further investment into these roles by government and willingness from academics to provide this form of support.

These resources could be provided free of charge to parents and would require only small periods of time to read [75]. The question of how this would be funded is clearly still an open one. However, the current situation, with increasing cyber attacks occurring [76], and children being poorly protected when online, cannot continue. Funding support workers would be a cost-effective way of addressing the situation.

### 3.1. Gauging Acceptability of Proposal

Given that we are proposing an intervention that is, as yet, unvalidated, we carried out a study that assesses subjectivity to help us understand people’s thinking with respect to the acceptability of this source of assistance. We deployed Q-methodology to carry out this final study. This method was introduced by Stephenson [77]. It specifically supports the systematic study of subjectivity. Q-methodology measures beliefs *as cultural phenomena*. and helps to reveal beliefs shared by groups of individuals. The findings from a Q-methodology analysis helps to assess the *nature* of subjectivity: ‘***what is the nature** of different groups’ thinking?*’, as opposed to ‘***how** are people thinking on the topic?*’. This methodology does not require large numbers of participants [78].

Q-methodology reveals correlations between subjects across a sample of variables: the “Q-set”. It is composed of ‘Q-statements’. A factor analysis is then used to isolate the most influential “factors”, which reflect cultural ways of thinking. The method’s strength is that it applies sophisticated factor analysis to support a qualitative analysis. The qualitative part of the analysis uses free-text responses where respondents are asked to explain why they agree or disagree with particular statements. This methodology does not seek to confirm or deny specific hypotheses; it aims to provide a sense of ‘*potentially complex and socially contested*’ issues [79]. Figure 3 depicts the steps involved in a Q-sort.

#### 3.1.1. Q Statements

Q-Statements for this study were derived from the results reported by [22] in response to questions relating to sources of help and the respondents’ free text responses. These were confirmed from the cybersecurity-related research literature.

Participants sort Q-Statements into a fixed quasi-normal distribution, ranging from −4 (disagree) to +4 (agree). They can amend and confirm their rankings and then provide open-ended comments for the most agreed-with (ranked +4) and most disagreed-with (ranked −4) statements (Table 2). This helps us, as researchers, to gain insights into the range of opinions about our topic of investigation [78].

#### 3.1.2. Recruiting

UK-based parents with children aged less than 18 years of age were recruited via the Prolific platform. Ethical approval was granted by the second author’s institution’s ethical review board. Measures to obtain informed consent and to ensure anonymity of participants’ responses were implemented.

Forty participants were recruited on the Prolific platform. This is consistent with recommended participant group sizes in Q-methodology [79]. 15 of the participants were female, 19 were male and 6 preferred not to specify their gender. Ages ranged from 25 to 74 years of age. Based on the pilot study timings, we paid participants £2 for 10 min of labour, exceeding the UK minimum wage. Participants did not provide any personal data, ensuring that participation was anonymous.

### 3.2. Analysis & Findings

We extracted factors using the principal component extraction technique and applied a varimax procedure for factor rotation. Factors with an eigenvalue in excess of 2.00, and having at least two significantly loading participants, were selected for interpretation (as recommended by [79]) (Figure A1 in the Appendix A).

Before we discuss each of the factors, we need to point out that all respondents most strongly disagreed with Statement 21: “*Hackers are not targeting children online so they don’t need to learn about cybersecurity until they are adults*”. This means that all parents are well aware of the risks to themselves and their children online. The final Q-Sorts are shown in Figure A2, Figure A3, Figure A4, Figure A5, Figure A6 in the Appendix A).


**Factor 1: The Government should enforce higher cybersecurity standards and they would accept cybersecurity-related advice.**


This factor explained 40% of the variance, with an eigenvalue of 15.83. 15 Participants (8M, 4F, 3 preferred not to say) belonged to this group, aged 25 to 74 (average age 43.6). One respondent said: “*They need to make sure we are safe in this technology used*” and “*i think they shoud care more about it, as no one is safe at the moment*”. However, they, themselves, struggled with cybersecurity knowledge, because they were looking for advice: agreement with statements 6, 8, and 9. One respondent said: “*Because I don’t have a lot of confidence in my knowledge of cyber security I trust the school and their advice and know they are thing me age appropriate cyber security. This is a definite agree that I would follow their advice*”.


**Factor 2: Embrace the idea of taking personal responsibility, and would accept advice from various sources.**


This factor explained 11% of the variance. Seven participants (4M, 2F, 1 preferred not to say) loaded to this factor, aged from 28 to 54 (average 41.2). This group embrace responsibility: “*i think they [think] it is the patents [sic] responsibility to teach their kids how to be safe*”, and “*I teach my children how to take responsibility for all their own actions and support them in doing so. I never make my children solely responsible*”. They also agreed with statements related to accepting advice from academics and family support workers. They do not believe that government has a role to play in this respect: “*It’s not up to the government to take responsibility. The internet is a worldwide channel. People will always find a way round*”, but they do believe that government-provided advice is helpful (agreeing with Statement 7).


**Factor 3: Do not trust the government to keep their children safe, and would accept advice from various other sources:**


This factor explained 8% of the variance. Eight participants (3M, 4F, 1 preferred not to say) loaded to this factor, aged from 31 to 49 (average 37). One respondent said: “*Because I don’t have any confidence in them for anything. Especially with everything else that is happening*”, but agreed with statements 6 and 8, indicating a willingness to accept and act upon cybersecurity advice, but not from the government (disagreeing with statement 7). It is safe to conclude that this group would like to take care of their own children’s cybersecurity educational needs, but do not want the government to be involved.


**Factor 4: Express a need for more cybersecurity advice and feel confident in acting upon advice:**


This factor explained 6% of the variance. Eight participants (3M, 5F) loaded to this factor, aged from 36 to 69 (average 41.2). This group also agree with statements 6, 8 and 9, these being most strongly agreed with. They have confidence to act upon this advice: “*I am very computer and cyber savvy I use technology all the time so i’m quite confiident [sic]*”. They would not blame themselves if their child was hacked (Statement 17).


**Factor 5: Believe that the government ought to require technology providers to do more, and are confident in their own ability to find cybersecurity guidance.**


This factor explained 5% of the variance. Two participants (1M, 1 preferred not to say) loaded to this factor, aged 35 and 62. This group does not have confidence in the government’s ability to teach children about cybersecurity best practice: “*Because I don’t have any confidence in them for anything. Especially with everything else that is happening*”. They had confidence in their own abilities, disagreeing with: “*I do not know that much about it myself to help*” and agreeing with statements 10 and 23.

## 4. Discussion and Reflection

The findings from our study confirm that participating UK parents embraced being responsibilized to educate their children about cybersecurity, probably a logical extension of all their other parental duties. As such, they accept this responsibilization, and do their best to find information to meet its demands.

It is thus unsurprising that three of the five factors strongly suggest that they could blame themselves if their children were hacked (Statement 17). In the free-text comments, one said: “*i feel children should mainly be taught cybersecurity by parents*”. Another said: “*It’s my responsibility upon this to keep my children safe*”. This confirms that UK parents are seeking more advice and guidance from external sources. The factors also pointed to a lack of trust in the UK government to provide this advice and guidance. This showed that parents were least likely to accept advice from the UK government. This validates the need to find a different way to provide them with the advice they need, as proposed by our intervention (Figure 2).

However, it is important to ensure that these support workers are given the knowledge they need to fulfil this new role, and it seems that cybersecurity academics are best placed to train them. Academics are trusted by parents—four of the five factors showing that people agreed that academics would have the most up to date information. Knowing that they are providing the information might increase the trustworthiness of the support workers in providing support and advice to parents.

The study also revealed that parents feel the government should force companies and technology providers to take their cybersecurity and cybersafety responsibilities more seriously. At the time of the study this was an issue in the UK press regarding the proposed Online Safety Bill [91]. This indicates that parents are willing to have some of the burden of responsibilization lifted from them.

### 4.1. Research Implications

Firstly, it would be interesting to understand why it is that parents so willingly accept their responsibilization to educate their children about cybersecurity, given the relative complexity of the risk management activities in this domain. They willingly leave their children’s education with respect to reading, writing, arithmetic and other areas to schools. They take their children to specialists to teach them to swim, play musical instruments and do gymnastics. Is it because cybersecurity is seen as an extension of cybersafety? Is it because they have become used to doing it? It would be interesting to explore this in greater depth by running focus groups with parents.

Secondly, the devil being in the details, we would have to flesh out the intervention to see exactly what would be required to implement it. It might be that the school liaison officers would be too intimidated by cybersecurity themselves. The government might not have the resources to fund this, especially given the current era of cost-of-living challenges [92]. Hence, fleshing out the intervention very carefully and coming up with funding models would be essential to determine its feasibility.

Thirdly, we need to understand what other barriers exist that may prevent parents being willing or able to take this advice on board.

### 4.2. Limitations

Our study was carried out while the UK’s Prime Minister election was ongoing, in the aftermath of politician misbehaviour which is bound to affect trust in government [93]. It is likely that these events shook confidence in the UK government’s competence, integrity and benevolence, all of which are essential in fostering trust in governments [94]. Our studies did not examine ethnicity. Work in other areas of responsibilization of education (for example during the lockdowns of 2020 and 2021) demonstrated that those from ethnic minorities were affected more negatively than others [95].

We carried out this study to identify the most acceptable and trustworthy source of correct cybersecurity advice, from parents’ perspective. The sample was relatively small, although comparable with other Q-methodology studies, and indeed in line with expert recommendations [77]. Even so, we would have to follow up with a large scale survey of the UK population in order to confirm our findings. Admittedly, as acknowledged in this paper, everything hinges on the UK government or charities being willing to provide funds for more support workers to fulfil this additional role of cyber adviser.

## 5. Conclusions and Future Work

The burden for citizens of managing their own cybersecurity practices and protections has been further placed on parents when it comes to handling their children’s cybersecurity education. However, traditional sources of advice and support (books, online sources, etc.) are unreliable, or not trusted (government). Hence, we have suggested a mechanism for providing parents with the support they need, which is: (1) well informed because they work with cyber security academics, and (2) already trusted by parents.

Our study confirmed that our proposed intervention for providing cybersecurity-related advice and support is likely to be accepted by UK parents, although we would have to carry out a more extensive study to confirm this. The benefit is likely to be well informed and confident parents in the cyber realm, and less vulnerable children across the United Kingdom.

## Figures and Tables

**Figure 1 children-10-01130-f001:**
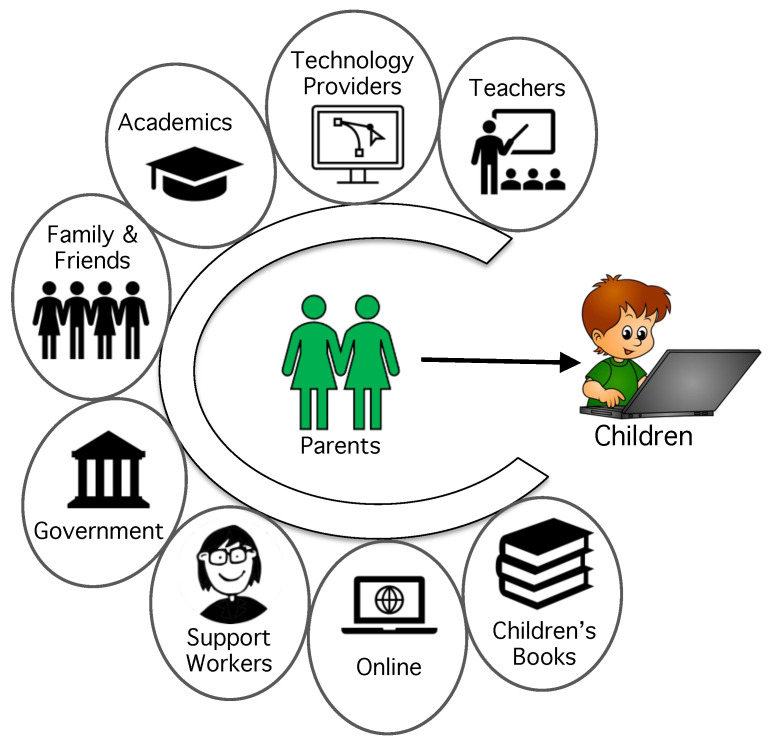
Possible Sources of Cyber Advice for Parents.

**Figure 2 children-10-01130-f002:**
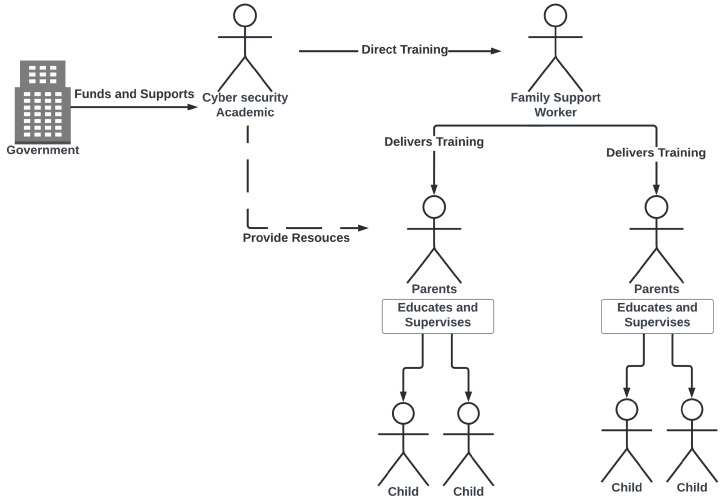
Proposed Support for Parent responsibilization.

**Figure 3 children-10-01130-f003:**
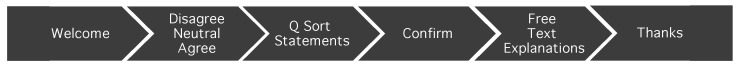
Q-Sorting Process.

**Table 1 children-10-01130-t001:** Summary of Advice Source Advice Provision Dimensions. (✔: Correct Advice, ✗: Incorrect Advice, ?: Currently Unknown, ≈: some correct/some incorrect).

Acronym	Advice Source	Provides Correct Advice	Trustworthiness
G	UK Government	✔	✗ (Mistrusted)
T	Teachers	≈	✔ (Trusted)
FF	Family & Friends	≈	✔ (Trusted)
O	Online	✗	✔ (Trusted)
A	Cybersecurity Academics	✔	✔ (Trusted)
BK	Children’s Books	✗	✔ (Trusted)
SW	Support Workers	?	✔ (Trusted)

**Table 2 children-10-01130-t002:** Q Statements (Acronyms from Table 1) G = Gov, P = Parents; T = Teachers; A = Academics; SW = Support Workers; TP = Technology Providers; C = Children.

#	Statement	Entity
1.	I know where to get reliable cybersecurity advice [80]	P
2.	I think the UK government provides helpful cybersecurity advice to parents (testing source)	G
3.	I can act upon the cybersecurity advice I am given by the UK government [81]	G
4.	My child’s teacher is well informed about cybersecurity [46]	T
5.	Cybersecurity academics are aware of the latest cybersecurity precautions (testing source)	A
6.	I would be happy to accept cybersecurity advice from family support workers/family liaison officers [82]	SW
7.	The UK government provides helpful cybersecurity advice [22]	G
8.	I would be happy to accept cybersecurity advice from cybersecurity academics (testing source)	A
9.	When I receive cybersecurity advice from my child’s school, I act on it (testing source)	S
10.	I feel confident in educating my child(ren) about cybersecurity precautions and good practice [75]	P
11.	I am uncertain about different cybersecurity practices [83]	P
12.	I am happy with how technology providers currently assure my child’s cybersecurity [84]	P
13.	I think technology providers could be doing more to provide better cyber security controls for children [84]	TP
14.	The UK government needs to enforce higher cybersecurity standards from technology providers [84]	G, TP
15.	The UK government should not meddle with the way technology providers deal assure cybersecurity	G
16.	There are plenty of cybersecurity resources for parents, schools do not need to get involved [85,86]	P
17.	I would blame myself if my child’s cybersecurity was compromised [87]	P
18.	I think children need to take responsibility for their online cyber security actions (testing responsibility)	C
19.	I am not an expert, it is up to my child’s educators to teach them about cybersecurity [32]	T
20.	The UK government should take ultimate responsibility for protecting my child’s cybersecurity (testing responsibility)	G
21.	Hackers are not targeting children online so they don’t need to learn about cybersecurity until they are adults [88]	
22.	Parents should figure out how to teach their children about cybersecurity without help from anyone else (testing responsibility)	P
23.	I am confident in my ability to teach my children about cybersecurity [89]	P
24.	I do not have confidence in the UK government’s ability to help me teach my child about cybersecurity [90]	G
25.	Parents should be responsible for teaching their children about cybersecurity [22]	P

## Data Availability

Dataset available from https://rke.abertay.ac.uk/en/datasets/ (accessed on 24 June 2023).

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
