# Peer review of "Who Is Best Placed to Support Cyber Responsibilized UK Parents?"

_children, 2023, doi:10.3390/children10071130_

Round 1
Reviewer 1 Report
Thank you for the opportunity to review you paper. I have attached a pdf file with comments. I regret that from my perspective, the paper needs a major rewrite to address the assumptions and limitations. The ideas are interesting but the paper lacks a sense of recommendations can be implemented given current funding and service limitations. The sample size does not allow for the generalizability suggested. This is a preliminary study that opens up an area for research but as a foundation to make policy recommendations it is not sound.

Reviewer 2 Report
Did you go to the radio station to talk about this? I heard from the researcher that employed the comparative analysis on this topic in the UK and Canada. This is an excellent study and I have a few minor comments.
1. This paper talks about children. I wonder how we are - as adults and parents - ready to deal with online issues including cyberbullying, especially considering the introduction of new technology such as ChatGPT can enhance a great level of uncertainty. Is it reasonable to expect parents to learn about these materials - which can be a source of inequality as well? Parents with greater levels of socioeconomic/psychosocial resources are better equipped to be exposed to learning materials. What do we do to ensure equality? Or in this case, equality is not important?
2. How about professionals? Although you list a long list, teachers and parents are not essentially professionals in this. Psychologists, medical doctors, computer scientists? Do we need a new type of professional for this? How do parents/teachers play a role in this?
3. How does this apply to children who lack fundamental resources? For example, see this news. https://www.bbc.com/news/uk-england-london-64224529. How do we ensure that the approaches described in this study can apply to every child in the UK and beyond? One-size-fits-all approach?
Round 2
Reviewer 1 Report
Thank you for your attention to revisions. Making recommendations in the area of public policy is tricky. On the attached there are three notes where I think the revisions may need a bit more attention - they are not major

Author Response
Thanks to the reviewer for their helpful comments. We have addressed them